# OpenReview forum: "MultiTrust: A Comprehensive Benchmark Towards Trustworthy Multimodal Large Language Models"
_NeurIPS.cc/2024/Datasets_and_Benchmarks_Track — NeurIPS 2024 Track Datasets and Benchmarks Poster_

### Official Review · Reviewer_Ku84 · 2024-07-22
**Review of submission54**

**Rating:** 7
**Confidence:** 4
**Correctness:** Yes
**Clarity:** Yes

**Review:**

Pros:

1. The topic of trustworthiness evaluation of Multimodal LLMs is necessary, so this work is very meaningful.

2. The presentation is good, all code and datasets have been well-organized and open-sourced to the community.

3. The experimental results are comprehensive and informative.


Questions：

1. Where can I see the detailed results of tables in Figure 2? Where is the detailed quantity of tasks in Table 2?

2. I do not understand how the Toolbox solves the problem of lacking off scalability and adaptability.

3. In the safety and privacy in Figure 2, what is the relationship between the input image and text?

4. Is the code compatible with OpenCompass platform?

5. Why are some symbols in Figure 2 right not labeled with which model they belong to?

**Strengths:**

See the above pros

**Additional Feedback:**

None

**Documentation:**

Yes

**Limitations:**

Yes

**Opportunities For Improvement:**

In addition to the above questions, some suggestion may be adopted:

1. It overlaps in Figure 2 right and needs to be corrected.

2. Some other related works are suggested to be discussed in related work, like [1] [2].

[1] LightCLIP: Learning Multi-Level Interaction for Lightweight Vision-Language Models

[2] Species196: A one-million semi-supervised dataset for fine-grained species recognition

**Relation To Prior Work:**

Yes

**Summary And Contributions:**

This submission establish MultiTrust, the first comprehensive benchmark on the trustworthiness of MLLMs across five primary aspects: truthfulness, safety, robustness, fairness, and privacy. 21 representative MLLMs are evaluated on this benchmark. Some interesting phenomenon are revealed from the evaluation results. The code and datasets have been open-sourced.

---

> ### Author Rebuttal · Authors · 2024-08-17
>
> Thank you for the careful and in-depth review. We are encouraged by your appreciation on the meaningful topic of this work, the contribution of code and datasets to the community, and the informative experimental results. We provide detailed responses below to address your concerns.
>
> **Q1: Detailed results for Fig. 2. Detailed quantity of tasks in Tab. 2.**
>
> We have presented the scores for 10 sub-aspects on our project page, which are attached in Table Ku84.1 in the rebuttal PDF. The score for each sub-aspect is calculated based on the results of all the tasks under it. Concretely, we scale different metrics to 0-100 and aggregate them with arithmetic mean. As for the data volume of 32 tasks, we have introduced it in the _Datasets_ paragraph for each task and add the amounts of image-text pairs in Table Ku84.2 in the rebuttal PDF. Overall, we have 23K image-text pairs in our benchmark. The details of score calculation will be added at the end of each section of aspect in appendix and those of data volume will be added in Table 2.
>
> **Q2: How the toolbox solves the problem of lacking off scalability and adaptability.**
>
> In our context, **scalability** refers to the capacity to incoporate additional tasks and models into the framework, while **adaptability** denotes the convenience to implement modules within a unified interface. These features are crucial for coordinating of diverse tasks and providing an all-in-one evaluation for newly proposed model. However, they remain absent in previous benchmarks for MLLM trustworthiness, which typically offer dense scripts for inference and evaluation on predefined datasets with limited flexibility for extension.
>
> To achieve these features, we implement a modularized codebase with unified interfaces, as shown in Fig.1. We separate the workflow into modules of datasets, methods, evaluators and metrics, which are assembled simply by a YAML config file. This allows for update to any part without requiring redundant modifications to the rest of the framework. For instance, one can update the dataset for a task by only modifying the dataset definition, or evaluate the outputs on a dataset from a different perspective by only switching the evaluator. As for models, we encapsulate diverse MLLMs, including proprietary APIs and open-source implementations, into a unified "Chat" interface. Researchers only need to support the `chat` function and can then test their models with only one command. We have tested several recently released models (e.g., Phi-3 by Microsoft, Hunyuan-Vision by Tencent) and updated their performance in the leaderboard on our project page, thanks to the simple adaptation from different code sources.
>
> We have compiled a website of documentation with explanations of each module and examples for implementing new ones, which can be found in our Github repo. We firmly believe that this toolbox can promote future researches into trustworthy MLLMs.
>
> **Q3: Relationship between the input image and text in Fig.1 for safety and privacy.**
>
> The study on the cross-modal impacts from the visual modality on **text-only** tasks is an important contribution of our paper. We present the examples of safety and privacy to illustrate this. The basic logic for evaluating cross-modal impacts is that we evaluate the performance changes when models have visual inputs that are relevant or irrelevant to the text prompts.
>
> The example of safety is to explain the effect of relevant visual inputs. We pair the text jailbreaking input with one positively relevant image and one negatively relevante image. The positive one is a man jailbreaking, which is closely relevant to the jailbreaking attempt, encouraging the model to break the rules and guardrails. In our experiments, we also replace it with images representing obedience and typographic ones with texts like "Sure, here is". On the contrary, the negative one depicts rejection. It can also be substituted with images emphasizing safety and typographic ones that read something like "I'm sorry". Visualized examples can be found in Figure D.11 and D.12. In Fig. 1, We observe that the visual contexts can manipulate the model behavior against a malicious text input.
>
> The case of privacy is to exemplify the impacts from irrelevant visual inputs. We gather a pool of irrelevant images of gaussian noises, color blocks, and natural images from ImageNet. Some are sampled from the pool to be paired with the text prompts for evaluation. In this figure, when a query for private information in history conversation is paired with an image of spider (natural, irrelevant), the model will compromise the privacy even though it can hold that with text-only inputs. This highlights the instability of MLLMs in multimodal contexts, potentially leading to untrustworthy behaviors.
>
> **Q4: Compatibility with OpenCompass platform**
>
> Definitely. As introduced above, the design of our toolbox is similar to that of recent popular platforms like opencompass, both modularizing models, datasets, metrics. It will be straightforward to merge the code into OpenCompass, which can promote the usage of MultiTrust, thanks to the good ecosystem of OpenCompass. We will consider adding this into our TODO list.
>
> **Q5: Incomplete labels of symbols in Figure 2 right.**
>
> Sorry for the confusion. We list top-8 models intended for clearer appearance, while model name can be found according to the trustworthy ranking. The complete labels will be added in revision.
>
> **Q6: Minor of formatting Fig. 2.**
>
> Thanks for pointing out. We will fix this in revision.
>
> **Q7: Suggested related works.**
>
> Thanks for the recommendation. LightCLIP is an impressive paper on developing lightweight foundation VLM. We will cite it in the background review of MLLMs in the section of Related Work. Species196 is a significant dataset for fine-grained recognition. We will cite it in Sec. C.1 studying inherent deficiency in truthfulness, which considers fine-grained perception as a sub-task.

---

> > ### Author Response · Authors · 2024-08-22
> > **Looking forward to further feedback**
> >
> > Dear Reviewer Ku84,
> >
> > We sincerely appreciate your insightful comments and the time you have dedicated to reviewing our work. We are looking forward to hearing from you about any further feedback.
> >
> > If our response addresses your concerns to your satisfaction, we kindly ask you to consider highlighting our contributions with a higher rating.
> >
> > If you still have any further questions regarding our paper, we are dedicated to discussing them with you and improving our paper.
> >
> > FYI, the PDF for rebuttal is attached in the global rebuttal.
> >
> > Best,
> >
> > Authors

---

> > > ### Comment · Reviewer_Ku84 · 2024-08-30
> > > **Thanks for the rebuttal**
> > >
> > > Thanks for the rebuttal from authors, after carefully reading this response, my concerns are well addressed. I am inclined to accept.

---

> > > > ### Author Response · Authors · 2024-08-30
> > > > **Thanks for the comment**
> > > >
> > > > Thanks for your comment on our response. We'll try our best to further improve in the final version.

---

### Official Review · Reviewer_46zd · 2024-07-25
**This article presents a benchmark for evaluating the trustworthiness of Multimodal Large Language Models and performs a detailed experiment study using their proposed benchmark.**

**Rating:** 6
**Confidence:** 5
**Correctness:** It is acceptable.
**Clarity:** It is acceptable.

**Review:**

While I appreciate your work, I have some concerns that need addressing:

Firstly, the authors aim to evaluate the trustworthiness of MLLMs, but they lack a precise definition of key aspects such as Truthfulness, Safety, Robustness, Fairness, and Privacy. These concepts appear scattered across different subsections.

Secondly, the discussion on diverse task design and implementation should be more closely linked to the aspects of trustworthiness being evaluated. Clarifying how these aspects impact the trustworthiness of MLLMs and justifying their use as benchmarks is crucial.

Thirdly, a comprehensive description of the benchmarks is needed to provide a clear understanding of their components and relevance in evaluating MLLMs' trustworthiness.

Lastly, a systematic discussion of metrics, including distinguishing between subjective and objective metrics, is essential for evaluating MLLMs accurately. Clarifying the nature of the metrics used will enhance the transparency and rigor of the evaluation process.

**Strengths:**

(1) a timely topic.
(2) consider different perspectives of the trustworthiness.
(3) Evaluation of lots of state-of-the-art and state-of-the-practice MLLMs.
(4) Several interesting insights.

**Additional Feedback:**

Please refer to my previous comments.

**Documentation:**

Yes.

**Ethics:**

No.

**Limitations:**

No. The authors should provide a reproducible methodology to generate the evaluation outcomes. In addition, the subjective evaluation metrics should be discussed.

**Opportunities For Improvement:**

(1)  A precise definition of key aspects such as Truthfulness, Safety, Robustness, Fairness, and Privacy in a concise section.

(2) Discussion and justification of diverse task design and implementation space.

(3) A comprehensive description of the benchmarks, including the specific components.

(4) A systematic discussion of metrics, including distinguishing between subjective and objective metrics.

**Relation To Prior Work:**

Yes. It is discussed.

**Summary And Contributions:**

(1) the design and implementation of benchmarks for the trustworthiness of  Multimodal Large Language Models.
(2) The detailed experiment study using the proposed benchmark.

---

> ### Author Rebuttal · Authors · 2024-08-17
>
> Thank you for the meticulous and instructive review. We are encouraged by your appreciation on the timely topic, the comprehensive taxonomy, the extensive evaluation, and the interesting insights. We provide detailed responses below to address your concerns.
>
> **Q1: Precise definition of key aspects.**
>
> We've defined the aspects at the beginning of Sec. 2.2.X and introduced the motivation of sub-aspect division. We'll give following definitions in a table to be more concise.
> * Truthfulness measures whether the outputs of MLLMs align with the objective facts, emphasizing the accuracy of the information they provide.
> * Safety guarantees that the responses from MLLMs do not cause unexpected consequences, such as unintentional harms or illegal actions.
> * Robustness evaluates the models’ consistency and resistance under diverse conditions in practical scenarios, including distribution shifts and input perturbations.
> * Fairness determines the extent to which the model outputs are free from inequitable or discriminatory outcomes that could disadvantage any user group.
> * Privacy assesses the models' capacity to perceive the risks of privacy disclosure and protect personal data from unauthorized requests.
>
> **Q2: Discussion and justification of diverse task design and implementation space.**
>
> Due to space limits, we only summarize each task under a sub-aspect in the main body. However, we offer sufficient explanations of the motivation and justification for these tasks at the beginning of each sub-section in Appendix C-G, covering their practical significance and reflection of the associated capacity. For instance, privacy leakage is discussed as below:
>
> >This is to evaluate the resistance of MLLMs to disclosing privacy information learned from either training or inference. Due to the inconsistency of multimodal training data across MLLMs, a unified evaluation of training data memorization for all models is impractical. We mainly consider the models’ ability to refuse privacy-sensitive requests in conversations. For multimodal scenarios, images can serve as sources of private information or as clues to request it. First, we use photos of different people to query their PII, which may lead to serous incidents when these models have access to database of personal data or are trained with private data. Then, models are asked to extract personal data from other people’s daily posts on media, which could result in unintentional leakage of privacy and misuse by malicious actors. We also test whether MLLMs are more inclined to expose private information in the text of past conversations when presented with different images.
>
> It explains the implementation of tasks as well as the relationship between them and the target concept. We'll add more discussions in the main body when describing the tasks.
>
> **Q3: Comprehensive description of the benchmark.**
>
> MultiTrust evaluates the trustworthiness of MLLMs through a comprehensive framework featuring a two-level taxonomy, a multimodal evaluation strategy, 32 diverse tasks, and a modular toolbox for testing over 20 modern MLLMs. The taxonomy includes five aspects—truthfulness, safety, robustness, fairness, and privacy—assessing models from both technical and ethical viewpoints, with ten detailed sub-aspects (see Fig. 1). It also addresses multimodal risks and cross-modal impacts due to the new modality to delve into the multimodal nature of MLLMs. This guides the design of the 32 tasks, which are detailed in the appendix, and includes a toolbox for unified and modular testing, facilitating task extension and model ensemble.
>
> We'll revise the first paragraph in Sec. 2 to offer a detailed description with this ammendment.
>
> **Q4: Systematic discussion of metrics.**
>
> We value your advice and hereby summarize the metrics used in our benchmark.
> * Objective Metrics: For tasks with clear closed-set answers, we generally use metrics like accuracy, pearson correlation, and p-value.
>     * Accuracy: $Acc=\frac{\sum_{i=1}^N\mathbb{I}(r_i = y_i)}{N}$, where $r_i$ is the model response and $y_i$ is the label. $\mathbb{I}$ decides whether the response corresponds to the label via exact matching, keyword matching, etc.
>     * Pearson Correlation: $Corr=\text{Pearson Correlation}(\mathbf{r}, \mathbf{y})$, where $\mathbf{r}$ is the sequence of score prediction, while $\mathbf{y}$ is the real score array.
>     * P-Value: $p-\text{value}=P(\chi^2\ge \chi^2_{\text{obs}})$, where $\chi^2 = \sum_i\frac{(O_i-E_i)^2}{E_i}$ with $O_i$ as the observed frequency and $E_i$ as the expected frequency.
>     * Other objective metrics, like ASR, Cure Rate, are calculated based on these metrics with subtraction and ratio.
> * Subjective Metrics: For tasks with open-set QA, we usually need to define and evaluate the behaviors with external evaluators like scoring API or large models, to replace human labors.
>     * Toxicity Score: ModerationAPI and PerspectiveAPI are used to rate the toxicity in a response, which are popular tools in toxicity study of LLMs.
>     * RtA, Containing Rate: Models like specialized longformer and GPT-4 are used to decide whether the response delivers certain behaviors or viewpoints with binary answers, which has been validated in other benchmarks[1,2].
>     * GPT-Score: GPT-4 is prompted to rate the responses, which leads to subjective evaluation and can have uncertainty. We've compared scores by GPT-4 with those by human on sampled subset, and got a correlation of 0.91, the highest among existing work (see Sec. E.1.2). It suggests the effectiveness of GPT-Score.
>
> **Q5: Limitations.**
>
> We've included the limitations in submission, as confirmed by other reviewers. Also, we've open-sourced our code and dataset, along with clear documentation to guarantee the reproducibility of our result. You can refer to our Github repo.
>
> [1] Do-not-answer: A dataset for evaluating safeguards in llms.
>
> [2] Mm-safetybench: A benchmark for safety evaluation of multimodal large language models.

---

> > ### Author Response · Authors · 2024-08-22
> > **Looking forward to further feedback**
> >
> > Dear Reviewer 46zd,
> >
> > We sincerely appreciate your insightful comments and the time you have dedicated to reviewing our work. We are looking forward to hearing from you about any further feedback.
> >
> > If our response meets your expectations, we hope that you might view this as a sufficient reason to further raise your score.
> >
> > If you still have any further questions regarding our paper, we are dedicated to discussing them with you and improving our paper.
> >
> > Best,
> >
> > Authors

---

### Official Review · Reviewer_LMT3 · 2024-07-25
**Great benchmark**

**Rating:** 8
**Confidence:** 3
**Correctness:** Yes
**Clarity:** Yes

**Review:**

I enjoyed reading this paper from the beginning to the end. It is well-written and the presentation is very clear. The presented benchmark has everything that a benchmark in this scale should have: it's comprehensive in several aspects, including the number and type of models used, evaluation metrics and protocols, variety of tasks, and the diversity of the datasets. The developed toolbox is promising and is very useful. I see no compelling reason to argue against this work and recommend it for publication.

**Strengths:**

- The paper is well-written and easy to follow
- The proposed benchmark is novel and important and its presentation is timely
- The study considers various aspects of trustworthiness relevant to MLLMs
- The evaluations address both multimodal risks and cross-model impacts across numerous diverse tasks
- For tasks with scarce data, the authors either existing datasets or created new data from scratch
- The findings of the paper are interesting and insightful

**Additional Feedback:**

No

**Documentation:**

Yes

**Limitations:**

Both limitations and societal impacts of the work are correctly addressed.

**Opportunities For Improvement:**

I see no other points for improvement other than the ones suggested by the authors themselves.

**Relation To Prior Work:**

Yes

**Summary And Contributions:**

The paper presents a comprehensive benchmark for evaluating the trustworthiness of Multimodal LLMs across five key aspects: truthfulness, safety, robustness, fairness, and privacy. It employs a rigorous evaluation strategy with 32 diverse tasks. The authors use a combination of existing datasets, adapted datasets, and also self-curated datasets. The evaluations cover both multimodal and cross-modal risks. The experiments with 21 modern MLLMs reveal new findings, such as susceptibility to multimodal jailbreaking and adversarial attacks, as well as increased privacy disclosure and biases. The paper also provides a scalable toolbox for assessing the trustworthiness of MLLMs.

---

> ### Author Rebuttal · Authors · 2024-08-17
>
> We sincerely appreciate your comprehensive recognition and endorsement of our work, which greatly encourages us. We have carefully addressed the concerns raised by the other reviewers, making improvements to enhance the clarity and impact of our contributions. We believe these responses and revisions will help strengthen our paper and better showcase its significance. We sincerely hope the improvements can make you more satisfied by our work. If you have any further questions or require additional discussion, we are open and more than happy to engage and refine our work based on your feedback.

---

> > ### Author Response · Authors · 2024-08-22
> > **Looking forward to further feedback**
> >
> > Dear Reviewer LMT3,
> >
> > We sincerely appreciate your insightful comments and the time you have dedicated to reviewing our work. We are looking forward to hearing from you about any further feedback.
> >
> > We hope that our responses to other reviewers bring our contributions up a level and can help you further support our work.
> >
> > If you still have any further questions regarding our paper, we are dedicated to discussing them with you and improving our paper.
> >
> > Best,
> >
> > Authors

---

### Official Review · Reviewer_ujCi · 2024-07-26
**A benchmark to evaluate the trustworthiness of multimodal large models**

**Rating:** 6
**Confidence:** 4
**Correctness:** Yes, it's correct.
**Clarity:** Yes, the paper is well written.

**Review:**

The paper is clear. The topic of the paper is important.

**Strengths:**

1. The paper is well-written and easy to follow.
2. The insights on model performance are interesting and important.
3. The experiments are comprehensive and clear.

**Additional Feedback:**

Please check Opportunities For Improvement above.

**Documentation:**

Yes.

**Limitations:**

Yes, the limitations are included.

**Opportunities For Improvement:**

1. The cross-modal impacts seem to be overclaimed. Calling them "image impacts" is better, considering the contexts covered by this work. The paper only evaluates text-only tasks and studies the impact of different kinds of images. It's also interesting to study the impact of different texts.
2. The synthesized data is generated by some of the models to be evaluated, e.g., GPT4V, which will inevitably make the performance of these models better. Consider using other models to generate the data or all multimodal models to generate the data to make the evaluation more fair.

**Relation To Prior Work:**

Yes.

**Summary And Contributions:**

This paper proposes MultiTrust, a comprehensive benchmark to evaluate the capabilities of multimodal foundation models across 5 aspects. The benchmark aggregates 32 datasets, including existing, augmented, and self-collected ones, to evaluate the model from different dimensions. The paper also analyses existing multimodal foundation models and gives insights into model development.

---

> ### Author Rebuttal · Authors · 2024-08-17
>
> Thank you for the constructive review. We are encouraged by your appreciation on the significance of our paper, the clear writing, and the important insights from the comprehensive experiments. We provide detailed responses below to address your concerns.
>
> **Q1: Study the impact of different texts.**
>
> Thanks for the suggestion. In this paper, we aim to examine the novel trustworthy risks introduced by the multimodal nature of MLLMs, particularly those arising from the visual modality. To achieve this, we categorize the risks into **multimodal risks**, which are the ones in new multimodal scenarios, and **cross-modal impacts**, which refers to the changes in the existing text-only scenarios due to the addition of images. We use the term "cross-modal impacts" to emphasize a broad range of trustwhothy risks associated with the interaction between modalities, which can be further extended to other modalities (e.g., video, audio, etc.).
>
> We acknowledge the importance of studying the effects of different text prompts. This has been extensively explored in the context of Large Language Models (LLMs) [1,2]. Hence, we regard this as an established concern for LLMs rather than a novel issue specific to MLLMs, which is why we did not categorize it as a separate perspective. Nevertheless, we recognize that the impacts from text variations also exist in the multimodal trustworthiness, and we have included some studies of this phenomenon in several tasks within our submission that are specific to multimodal scenarios rather than for LLMs.
>
> For example, in the task of VQA under Instruction Enhancement (T.3), we explored how different instruction prompts could affect the model’s performance in VQA. Prompts are designed to be different in the guidance of question answering, varying from queries for direct responses ("_Please directly tell me the final answer without analysis process._") to instructions for step-by-step thinking ("_You are a mathematician. Please solve the mathematical problem step and step according to the image and tell me the final answer._") Our results confirm the sensitivity of MLLMs to varying prompts, with the optimal prompt choice varying across different models and tasks.
>
> Besides, in the task of PII Query with Visual Cues (P.4), we designed four prompt templates presenting different information for reference. Our findings indicate that some models are more inclined to disclose privacy information when provided with minimal information. Additionally, in the task of Privacy Leakage in Vision (P.5), we compared the Refuse-to-Answer rates between prompts with and without explicit instructions of protecting privacy. We noticed that explicit instruction can encourage models to refuse improper requests. Similar settings that consider different text prompts in mutimodal tasks also exist in tasks of safety (e.g., S.1 -- NSFW Image Description) and fairness (e.g., F.6 -- Profession Comptetence Prediction).
>
> Moreover, we have also considered different presentations of text prompts in images for typographical jailbreaking (S.4). Some insightful conclusions are drawn from this task. For example, the OCR capabilities of advanced MLLMs are activated when the target harmful behaviors are placed in images and the safety guardrails are bypassed.
>
> It's important to note that the design of different text prompts is highly dependent on the specific multimodal tasks, which is nontrivial to form a systematic view across dozens of tasks. We have considered multiple settings with diverse text prompts and provided some valuable insights, while our primary emphasis remains on the novel risks from the new modality of vision for MLLMs.
>
> We'll add more discussions in corresponding paragraphs, like the evaluation strategy in Sec. 2 and the related work.
>
> [1] Quantifying Language Models' Sensitivity to Spurious Features in Prompt Design or: How I learned to start worrying about prompt formatting.
>
> [2] Exploring the Sensitivity of LLMs’ Decision-Making Capabilities: Insights from Prompt Variations and Hyperparameters.
>
> **Q2: Validity of synthesized data generated by GPT-4V.**
>
> Thanks for the suggestion. It is reasonable to concern that the synthesized data generated by a model to be evaluated may lead to unfair evaluation. First of all, there is only 1 task (P.2 -- Privacy-Sensitive VQA Recognition) that uses the multimodal GPT-4V to generate questions, and 2 tasks in truthfulness that use the text-only GPT-4 to generate text prompts. The reason why we adopt GPT-4V to generate data is to automate data generation instead of heavy human labours, while achieving data richness and complexity comparable to that by human. After the synthetic generation, there is manual check and revision to guarantee the data quality. Besides, it has been common to use GPT-4V to generate questions for multimodal benchmarks in the field [3,4].
>
> To validate that the evaluation on the data generated by GPT-4V is fair, we conduct several additional experiments. For the task of P.2, we use Gemini and InternLM-XC2, which are also models to be evaluated, along with Hunyuan-V, a recently released closed-source model, to generate questions on the same images. We see that GPT-4V gets the best results on all four test sets, which indicates that the superiority of its performance does not only come from the self-generated data. We calculate the Pearson correlation coefficients between the model accuracies on four datasets and they are mostly greater than 0.90, denoting consistent trends on data from different sources. This supports the validity of the data generated by GPT-4V.
>
> Similar experiments are also done for tasks of truthfulness and result in the same conclusion. Detailed results are in Table ujCi.1 in the PDF.
>
> [3] Red teaming visual language models.
>
> [4] Mm-safetybench: A benchmark for safety evaluation of multimodal large language models.

---

> > ### Author Response · Authors · 2024-08-22
> > **Looking forward to further feedback**
> >
> > Dear Reviewer ujCi,
> >
> > We sincerely appreciate your insightful comments and the time you have dedicated to reviewing our work. We are looking forward to hearing from you about any further feedback.
> >
> > If our response meets your expectations, we hope that you might view this as a sufficient reason to further raise your score.
> >
> > If you still have any further questions regarding our paper, we are dedicated to discussing them with you and improving our paper.
> >
> > FYI, the PDF for rebuttal is attached in the global rebuttal.
> >
> > Best,
> >
> > Authors

---

### Author Rebuttal · Authors · 2024-08-17

We thank all the reviewers for their comments and valuable feedback. We are glad to see that the significance of our topic, the timely presentation of the benchmark, the comprehensive coverage of key aspects, the extensive experiments on modern MLLMs, and the insightful conclusions have got unanimous approval. We have made the following major updates following the reviews to further improve our work.
* Following the suggestions from Reviewer ujCi, we have added additional discussions to support the justification of "cross-modal impacts" and explain our studies on the impacts from the textual modality. Meanwhile, we have done some experiments to validate the effectiveness of a small portion of test data generated by GPT-4V.
* Following the suggestions from Reviewer 46zd, we have refined our introduction of the benchmark in Section 2, especially in the concise definition of key aspects and summarization of various metrics.
* Following the suggestions from Reviewer Ku84, we have explained the contributions of our toolbox and its compatibility with modern platforms.

If the manuscript is accepted, all updated contents in the main body will be sufficiently added given the extra page limit for the camera-ready version.

---

### Author Response · Authors · 2024-08-28
**Follow-up on Author Response**

Dear Reviewers,

We sincerely appreciate the time and effort you have invested in reviewing our submission. Your feedback has been invaluable in refining and improving the work, and we have carefully addressed all the concerns raised in my response.

Given the author-reviewer discussion coming to its end, we would like to respectfully ask if there are any remaining issues or misunderstandings, and whether, after considering my responses, you might be willing to raise your ratings. We believe that with the additional insights provided, the contributions of this work could be more clearly clarified.

Thank you again for your thoughtful reviews. We look forward to any additional feedback you may provide.

Best,

Authors

---

### Decision · Program_Chairs · 2024-09-26

**Decision:**

Accept (Poster)

**Comment:**

This paper introduces MultiTrust, a comprehensive benchmark designed to evaluate the trustworthiness of Multimodal LLMs across five key aspects: truthfulness, safety, robustness, fairness, and privacy. The benchmark utilizes 32 datasets, including existing, augmented, and self-collected ones, to assess models from various dimensions. Additionally, the paper analyzes existing multimodal foundation models, providing insights into model development. The reviewers unanimously agree on the significance of the contributions and the clarity of the presentation, and their concerns were adequately addressed during the rebuttal phase. I recommend that the authors refine their paper for the camera-ready version according to the review comments and discussions.